# From Water to Land: The Structural Construction and Molecular Switches in Lungs during Metamorphosis of *Microhyla fissipes*

**DOI:** 10.3390/biology11040528

**Published:** 2022-03-30

**Authors:** Liming Chang, Meihua Zhang, Qiheng Chen, Jiongyu Liu, Wei Zhu, Jianping Jiang

**Affiliations:** 1CAS Key Laboratory of Mountain Ecological Restoration and Bioresource Utilization & Ecological Restoration and Biodiversity Conservation Key Laboratory of Sichuan Province, Chengdu Institute of Biology, University of Chinese Academy of Sciences, Chengdu 610041, China; changlm@cib.ac.cn (L.C.); zhangmh@cib.ac.cn (M.Z.); chengqh1@cib.ac.cn (Q.C.); liujy@cib.ac.cn (J.L.); 2University of Chinese Academy of Sciences, Beijing 100049, China

**Keywords:** WGCNA, lung development, histological structure, molecular changes, amphibian

## Abstract

**Simple Summary:**

The functionalization of lungs is a necessity for most anurans to breathe on land. Previous studies have focused on the morphological and physiological functions of amphibian lungs, while the microstructural changes and molecular mechanisms that underpin the functional maturation of lungs remain under-researched. We used integrated histology and transcriptomics to study the critical cytological and molecular events associated with lung maturation in *Microhyla fissipes*. The results illuminated the molecular processes and their coordination in lung development, providing an insight into the transition of amphibians from aquatic to terrestrial life stages.

**Abstract:**

Most anurans must undergo metamorphosis to adapt to terrestrial life. This process enhances the air-breathing ability of the lungs to cope with the change in oxygen medium from water to air. Revealing the structural construction and molecular switches of lung organogenesis is essential to understanding the realization of the air-breathing function. In this study, histology and transcriptomics were conducted in combination to explore these issues in *Microhyla fissipes*’ lungs during metamorphosis. During the pro-metamorphic phase, histological structural improvement of the alveolar wall is accompanied by robust substrate metabolism and protein turnover. The lungs, at the metamorphic climax phase, are characterized by an increased number of cilia in the alveolar epithelial cells and collagenous fibers in the connective tissues, corresponding to the transcriptional upregulation of cilia and extracellular matrix-related genes. Post-metamorphic lungs strengthen their contracting function, as suggested by the thickened muscle layer and the upregulated expression of genes involved in muscle contraction. The blood–gas barrier is fully developed in adult lungs, the transcriptional features of which are tissue growth and regulation of differentiation and immunity. Importantly, significant transcriptional switches of pulmonary surfactant protein and hemoglobin facilitate air breathing. Our results illuminated four key steps of lung development for amphibians to transition from water to land.

## 1. Introduction

Metamorphosis is a critical process for most anurans to accomplish the transition from aquatic to terrestrial life [1,2]. During this process, numerous organ systems undergo rapid morphological and functional changes to adapt to new environments [3,4]. These include the resorption of larval-specific tissues, such as gills and tails; formation of adult tissues such as hindlimbs, forelimbs, and lower jaws; and, more commonly, remodeling of larval organs (e.g., gastrointestinal tract, central nervous system, skin, kidney, and liver) [5,6,7,8]. Thus, amphibian metamorphosis has long been a paradigm for the study of cell proliferation and differentiation, programmed cell death, and tissue remodeling. From an evolutionary perspective, the sequential molecular variations underlying these morphological and functional changes provide clues to understanding the historical adaptation of vertebrates to terrestrial environments [1,9,10].

Air breathing is a milestone event in the evolutionary history of vertebrates. Amphibians are the earliest tetrapods to adapt to the terrestrial environment, and their lungs play an indispensable role in enabling air respiration for most anuran species. As a result of their complicated life cycle, the metamorphosis of anuran larvae is accompanied by a shift in primary respiratory organs. Gills are responsible for breathing during the aquatic life stage; skin is involved in gas exchange during all anuran life stages, while lungs are responsible for gas exchange during the terrestrial life stage [1,11,12,13]. During metamorphosis, functionalization of the lung is most fascinating because it resolves the challenges of the oxygen medium changing from water to air, such as the changes in environmental toxicants, pathogens [14], oxidative stress, and reactive oxygen species [15,16]. Although the morphological and physiological functions of the amphibian lung have received much attention, the microstructural changes and molecular mechanisms that underpin the functional maturation of lungs remain to be illuminated. For example, in addition to alveolar architecture, developmental levels of the capillary network, muscle tissue, and immune system in the lungs can also determine the efficiency of gas exchange. However, the stepwise maturation process of these functions during metamorphosis has been overlooked. A systematic understanding of the structural construction and molecular switches in the lungs during metamorphosis will likely provide mechanistic insight into the stepwise adaptation of the amphibian respiratory system to air breathing.

Two groups of proteins, pulmonary surfactant-associated proteins (SFTPs) and hemoglobin (HB), play important roles in the maintenance of normal lung respiration. SFTPs are constitutive components of lung surfactant, with critical functions in reducing the surface tension of pulmonary alveoli and conferring innate immunity [17]. The content of surfactant is an important indicator of lung function. HB is the carrier of oxygen and carbon dioxide in animals. Although erythrocytes are commonly not considered to be a component of lungs, they are functionally impartible from the respiratory organs, and lung maturation is accompanied by an incremental increase in blood flux through the lungs. Thus, to some extent, quantifying the transcriptional levels of HB in lungs can reflect their respiratory efficiency [18].

Although functional lungs have been detected in most anuran species at their adult phase, their contributions to total respiration requirements vary among species. *Xenopus laevis* is the most familiar amphibian model for developmental studies. This animal remains in an aquatic environment after metamorphosis, and correspondingly, its lungs are structurally simple, and present as oval sacs surrounded by a capillary net [19,20,21]. For most other frogs and toads (e.g., *Pelophylax nigromaculatus* and *Bufo vulgaris*) that live in a terrestrial environment after metamorphosis, the lungs are more complicated in morphology and significant in function, and, correspondingly, they are finely divided into many respiratory spaces and copiously distributed capillary networks [19]. These amphibians, with a distinct aquatic and terrestrial life history, are more appropriate for studying the molecular basis of lung functionalization during metamorphosis.

*Microhyla fissipes* (Anura: Microhylidae) is a very good model for exploring anuran metamorphosis, owing to its representativeness as a land-dwelling species at the adult phase, its superiority in development rate and transparency of tadpole skin [10,22], and because we have its complete embryonic developmental table [23]. Here, integrated histology (histological section and transmission electron microscopy) and transcriptomics were conducted to study the critical cytological and molecular events associated with lung maturation during the pro-metamorphic (Stages 37–41), metamorphic climax (Stage 43), post-metamorphic (Stage 45), and adult phases. Then, we explored the developmental patterns by which the different cellular functions mature, as well as the metabolic processes that support the morphogenesis. This study highlights the role of lung development during metamorphosis in facilitating the transition of amphibians from water to land.

## 2. Materials and Methods

### 2.1. Animal and Daily Culture

The *M. fissipes* adults were collected from farmlands (E 103.459885°, N 30.744614°, 701 m) located in Shifang City, Sichuan Province, China, 15 July 2019. Adult male and female frogs were bred following standard procedures for artificially induced spawning [23] and egg production. Four egg clutches (ranging from 200 to 500 eggs) of *M. fissipes* were collected and placed into 12 aquatic containers (length 42 × width 30 × depth 10 cm, water depth = 5 cm) and hatched at 25 ± 0.5 °C (water temperature, light/dark = 12:12 h). The hatched larvae were fed a solution of boiled chicken egg yolk once a day for 2 days. Then, tadpoles were fed spirulina powder (China National Salt Industry Corporation, Beijing, China) once a day, and water was replaced every 2 days. The developmental stages of tadpoles were identified according to the staging table reported by Wang et al. (2017) [23].

### 2.2. Experimental Design and Sampling

This experiment was conducted to investigate amphibian lung development from the aquatic life stage to the terrestrial life stage. *M. fissipes* individuals were collected during the pro-metamorphic (Stages 37, 39, and 41), metamorphic climax (Stage 43), post-metamorphic (Stage 45), and adult phases (Figure 1A). After individuals were euthanized with Tricaine (MS-222), the lung tissues were collected for micro-computed tomography (Micro-CT, *n* = 2 each stage), histological sections (both light and electron microscopy, *n* = 3 each stage), and RNA-seq (*n* = 3 each stage). The animal procedures were approved by the Animal Care and Use Committee of Chengdu Institute of Biology (permit no. 2015-AR-JJP-01).

### 2.3. Micro-CT

After anesthetization with MS-222, tadpoles were fixed in 4% paraformaldehyde for 1 h. All specimens were scanned using a scanner (Quantum GXmicro-CT Imaging System, Per-kinElmer, Waltham, MA, USA) at the herpetology lab, Chengdu Institute of Biology, Chinese Academy of Sciences, with the following parameters: scanning current, 70 eV; 10 μM; field of view: 36 × 36 mm for acquisition, 25 × 25 mm for reconstruction; scan duration, 15 min. Scanning images of the lungs were segmented and color-rendered using the 3D software, Materialise Mimics (Materialise’s interactive medical image control system, Materialise Companies, Leuven, Belgium).

### 2.4. Histological Analysis

The method of histological sectioning followed the protocol described by Chang et al. (2021) [24]. In brief, histological sections were stained with Masson trichrome stain for collagen fibers and myofibers. Section staining followed the instructions of the commercial kits purchased from Servicebio Technology Co., Ltd. (Wuhan, China). After sealing with resinene, the histological sections were analyzed using a Nikon E200 microscope equipped with an industrial digital camera (APTINA CMOS Sensor, San Jose, CA, USA). For each stage, three biological replicants were conducted.

### 2.5. Transmission Electron Microscopic (TEM) Observation

Fresh lung tissues were collected and cut into small (1 mm^3^) blocks. After they were fixed in 3% glutaraldehyde (6 h, 4 °C), the tissue blocks were rinsed in 0.1 M Sorensen’s phosphate buffer (pH 7.4) three times (15 min each time), and postfixed for 2 h in 1% osmium tetroxide in the same buffer. The tissue blocks were rinsed again and dehydrated using a graded series of ethanol (30, 50, 70, 80, 95, and 100%, 20 min for each grade). After penetration with a mixture of acetone and EMBed 812 overnight at 37 °C, tissue blocks were embedded in EMBed 812. The embedding models with resin and samples were put into a 65 °C oven to polymerize for more than 48 h. The resin blocks were cut into 60–80 nm ultrathin sections using an ultramicrotome (Leica UC7) and Diamond slicer (Daitome Ultra 45°). Ultrathin sections were fished out onto the 150-mesh cuprum grids with formvar film and stained in 2% uranium acetate saturated alcohol solution for 8 min. After rinsing with 70% ethanol and ultrapure water, the ultrathin sections were stained in 2.6% lead citrate for 8 min. After drying using filter paper, the cuprum grids were put into the grid board and dried overnight at room temperature. The cuprum grids were observed under TEM (Hitachi, HT7800/HT7700) operating at 60 kV and photographed using a CCD digital camera.

### 2.6. Transcriptomic Analyses

In order to meet the minimum tissue amount for RNA extraction, 30, 30, 20, 15, and 10 individual lungs were merged as one sample at Stages 37, 39, 41, 43, and 45, respectively; three samples were prepared for each phase as biological repetitions (*n* = 3 each stage). For adults (*n* = 3 each stage), the lungs of two males and one female were collected, and each lung was used as one sample. The samples were fresh frozen in liquid nitrogen and stored in a −80 °C freezer until RNA extraction. Total RNA extraction followed the protocol of TRIzol (Life Technologies Corp., Carlsbad, CA, USA). For each sample, 1 μg RNA was used for library construction with a NEBNext^®^Ultra™ RNA Library Prep Kit for Illumina^®^ (NEB, Ipswich, MA, USA). Sequencing was conducted on an Illumina Hiseq 2000 platform from Biomarker Technologies Co., Ltd. (Rohnert Park, CA, USA) and paired-end reads were generated. The sequencing data in this study were submitted to the Genome Sequence Archive (GSA, https://bigd.big.ac.cn/gsa/ (accessed on 1 January 2022)) under accession number PRJCA004230. Clean data were obtained by removing reads from the raw data that contained adapter and ploy-N, and low-quality reads. The transcriptome was assembled with Trinity [25] and annotated by querying against NR (NCBI non-redundant protein sequences), Pfam (protein family), KOG/COG/eggNOG (clusters of orthologous groups of proteins), Swiss-Prot (a manually annotated and reviewed protein sequence database), KEGG (Kyoto Encyclopedia of Genes and Genomes), and GO (Gene Ontology). Gene expression levels were estimated by RSEM [26]. The differential expression analyses between developmental stages were performed using the DESeq2 based on negative binomial distribution [27]. Differentially expressed genes (DEGs) should meet q < 0.05 after Benjamini and Hochberg’s correction.

### 2.7. Weighted Correlation Network Analysis (WGCNA)

The gene clusters or modules potentially associated with each developmental stage were screened out by WGCNA [28]. WGCNA is a systematic biological method used to construct a scale-free network based on gene expression profiles. First, we constructed a similarity matrix that calculated the absolute value of the Pearson’s correlation coefficient between two genes based on expression data. Then, the similarity matrix was converted into an adjacency matrix, where the β value was the soft threshold (power value) to enhance strong connections and to disregard weak correlations between genes in the adjacency matrix. Next, the adjacency matrix was converted into a topological matrix (TOM) to describe the association strength between the genes. The TOM was used as the input for hierarchical clustering analysis of genes, and the dynamic tree cut algorithm was applied to identify network modules. The most representative genes, module eigengenes (MEs), were the first principal components, representing the overall level of gene expression in individual modules. Module membership (MM) was measured using Pearson’s correlation coefficient of the expression profile of one gene in all samples and one ME. Lastly, gene significance (GS) value was used to evaluate the gene with developmental stage information (Appendix A in Appendix A). The higher the value of GS, the better representativeness it holds for a specific developmental stage. Thus, the expression profile of DEGs was used to construct a free-scale network and identify significant modules related to developmental stage traits. We screened the core genes (GS > 0.7 and MM > 0.7) to reveal the biological function of modules.

## 3. Results

### 3.1. Structure Observations

The lungs of *M. fissipes* increased in volume and became complicated in morphological and histological structure with the proceeding of metamorphosis (Figure 1B). The pro-metamorphic lung (S37–41) presented a simple sac-like structure, which comprised a single layer of epithelium surrounded by a thin layer of mesenchyme. The most prominent feature was the thickening of the alveolar wall during this stage. During the metamorphic climax phase, increasing alveolar septa separated the lung into several irregular compartments. Histologically, there was an obvious increment in the number of collagenous fibers at the mesenchyme layer, which were stained in blue by Masson’s trichrome method (Figure 1C). At the post-metamorphosis phase, the number of collagenous fibers in the mesenchyme layer decreased dramatically. Instead, a thin but integral smooth muscle layer (stained in red) was formed in the lungs (Figure 1C). In adults, the lung wall was characterized by the formation of septa, which were rich in pulmonary blood vessels and a series of peripheral alveolar sacs originated from these septa. The electron microscopy results suggested that the alveolar epithelial cells appeared to be oval-like during the pro-metamorphic phase and metamorphic climax phase, and then matured as squamous cells with irregular shapes during the post-metamorphosis and adult phases (Figure 1D). With the proceeding of metamorphosis, the cilia of alveolar epithelial cells grew in number and length, especially after the metamorphic climax (S43). Blood capillaries across the alveolar wall increased in number, resulting in reduced distances between alveolar epithelial and endothelial cells. Notably, the blood endothelial cells were surrounded by large alveolar epithelial cells in the adult lungs, which suggested a mature blood–gas barrier (Figure 1D). Structurally, the lungs showed stepwise maturation within the four developmental phases.

### 3.2. Functional Analysis of Co-Expressed Modules

In this study, a total of 11,641 genes were identified as differently expressed between stages (pairwise comparisons) (Figure 2A). WGCNA was conducted for these DEGs, and they were classified into 13 gene clusters/modules according to their expression patterns across stages. Among the 13 clusters, 3 modules, 1 module, 1 module, and 1 module showed significant associations (R^2^ > 0.7 and *p* < 0.001) with pro-metamorphic (S37, 39, and 41), metamorphic climax (S43), post-metamorphic (S45), and adult phases, respectively (Figure 2B). Therefore, these gene modules featured the corresponding phases.

The featured gene modules of different developmental phases were queried against the KEGG and Reactome databases for functional enrichment analyses (Figure 2C). The results showed that the featured gene modules of the pro-metamorphic phase were enriched in substance metabolism and immunity. During the metamorphosis climax phase, the featured gene modules highlighted ECM remodeling (i.e., ECM organization, collagen formation, and signaling by MET) and ciliogenesis. The featured gene modules of post-metamorphic lungs highlighted muscle contraction, substance metabolism, and energy metabolism. The featured gene modules of adult lungs were enriched in substance metabolism, regulation of proliferation and differentiation, pulmonary circulation, and immunity.

Furthermore, we explored the variation patterns of featured genes at different developmental phases. In the pro-metamorphic phase, we identified 23, 15, 17, and 25 featured genes involved in protein turnover (i.e., ribosomal component and proteolysis-related genes), lipid metabolism, amino acid metabolism, and carbohydrate metabolism, respectively (Figure 3). Their expression in the lungs was maintained at relatively high levels at the pro-metamorphic phase (S37, S39, and S41), and then downregulated after the metamorphic climax phase. In the metamorphic climax phase, the featured genes included 22 and 6 genes involved in ECM and cilium construction, respectively. Their expression levels peaked at the metamorphic climax phase (Figure 4). Noticeably, there were six mitotic genes that belonged to this gene module (Figure 4). The featured genes of post-metamorphic lungs comprised 11, 21, and 28 genes enriched for substrate metabolism, energy metabolism (i.e., ATP synthase, NADH dehydrogenase, and ADP/ATP translocase), and muscle contraction (i.e., tropomyosin, myosin, filamin, and myozenin), respectively. These genes had the highest transcription during the post-metamorphic phase (Figure 5). A total of 29, 16, and 14 featured genes of adult lungs were involved in immunity, metabolism, and tissue growth and differentiation regulation-related pathways (i.e., Wnt and Notch signaling pathways). These genes showed gradually upregulated transcription in the lungs with the maturation of *M. fissipes* (Figure 6).

### 3.3. Transcriptional Switches of Respiration Functional Proteins

A total of five SFTP transcripts were identified in this study—SFTPA1-like, SFTPA2-like, SFTPB, SFTPC, and SFTPD-like—and they presented different patterns with the stages (Figure 7). The transcriptional levels of SFTPA1-like and SFTPB increased as metamorphosis proceeded and peaked in adult lungs. The transcription of SFTPC dropped to the lowest level during the metamorphic climax phase, and then increased with the maturation of lungs. The transcription of SFTPD-like and SFTPA2-like peaked at Stage 41, and then decreased to an undetectable level during the metamorphic climax and adult phases, respectively.

There were seven transcripts of HB subunits identified in this study, and they presented different patterns with the stages (Figure 7). The transcriptional levels of HBα3, HBε and larval β globin peaked in the late pro-metamorphic phase (Stage 41), and then decreased dramatically during the metamorphic climax phase, and maintained extreme low levels at the terrestrial life stage. In contrast, the transcriptional levels of HBα5, HBα-B-like, HBα-C-like, and HBβ2 were maintained at extremely low levels during the pro-metamorphic phase and then sharply upregulated during the metamorphic climax phase and peaked in adult lungs.

## 4. Discussion

### 4.1. Rapid Growth and Robust Substrate Metabolism of Lungs during the Pro-Metamorphic Phase

The pro-metamorphic phase is a thoroughly aquatic life stage of *M. fissipes* larvae. During this phase, a structurally simple blood–gas barrier suggests that lung function is immature. However, the pro-metamorphosis phase is devoted to rapid somatic growth for tadpoles [23]. Rapid growth in size occurred during the pro-metamorphosis phase. Consistent with these morphological changes, the lungs of pro-metamorphic larvae showed robust transcription of genes in protein turnover, substrate metabolism, and surfactant metabolism (Figure 3). Protein turnover, which refers to protein synthesis and breakdown, is always extensive during rapid cell growth [29]. As growth rates increase, both whole-animal protein synthesis and degradation rates increase linearly; growth occurs because of the preponderance of synthesis over degradation [30]. Protein turnover is the most energy-intensive process in cells, and thus, accelerated protein metabolism relies on coupled upregulation of substrate metabolism, including lipid, amino acid, and carbohydrate metabolism (Figure 3). Accordingly, robust substrate metabolism might be a key event in facilitating lung growth during the pro-metamorphic phase.

### 4.2. Construction of Extracellular Matrix and Accelerated Development of Lungs during the Metamorphic Phase

Amphibian larvae begin to transform from an aquatic environment to a juvenile terrestrial environment during the metamorphic climax phase. This means that their lungs begin to morphologically and functionally mature during this phase. The lungs of *M. fissipes* larvae underwent dramatic morphological changes. Extracellular collagenous fibers, or other ECM components, are significant in directing rapid organogenesis in developing animals [31]. During the process, the ECM interacts with cells to regulate diverse functions, including proliferation, migration, and differentiation [32]. Consistent with histological observations, metamorphic lungs showed transcriptional upregulation of genes involved in ECM and cilium construction (Figure 2C and Figure 4). The ECM is a highly dynamic structure that is present in all tissues, particularly in developing organs [32,33]. In our study, ECM-related genes, whose translation peaked during the metamorphic climax phase, included matrix metalloproteinases (MMPs), elastin, and collagen (Figure 4). MMPs are the main enzymes involved in ECM remodeling [34,35]. Their proteolytic actions are crucial to organogenesis and branching morphogenesis. Elastin and collagen are constitutive components of ECM, responsible for ECM stiffness in neonatal lungs and mediating the LRP5/TIE2 pathway to promote lung development in mice [36,37]. In addition to developments in ECM, our results suggested that metamorphic lungs had more developed cilia in their alveolar epithelial cells (Figure 1D), and upregulation of cilium-related genes was another major transcriptional feature of metamorphic lungs (Figure 4). Cilia are an important structural basis for lung alveolar epithelial cells to increase the area of gas exchange, holding a protective coat of mucus to the epithelium and facilitating the cellular secretory function. It is worth mentioning that the mitotic genes showed prominent transcriptional upregulation (Figure 4). This suggested coupled ECM construction and cell proliferation of lung morphogenesis during the metamorphic climax phase.

### 4.3. Reinforcing of Muscle Function in Post-Metamorphic Lungs

During the post-metamorphic phase, *M. fissipes* rely on air breathing completely. The lungs during this phase have a relatively complicated structure. The most prominent morphological improvement is increased smooth muscle fibers in the lung during this phase (Figure 2). Ancestrally, smooth muscle was likely involved in tension regulation and may be involved in deflation, and functions largely to maintain the internal lung structure in amphibians [38]. In support of our morphological observations, the major translational feature during this phase was robust expression of genes involved in muscle function and energy metabolism (Figure 2C and Figure 5). These genes included tropomyosin, myosin, filamin, and myozenin, which are constitutive components of muscle fibers [39]. Meanwhile, enhanced substrate metabolism and energy metabolism might provide the energy to support smooth muscle contraction during the respiratory process. This means that capacity for active respiration is likely improved comprehensively during this phase.

### 4.4. Transcriptional Switches of Respiration Functional Proteins during the Metamorphic Climax Phase

Pulmonary surfactant is a lipoprotein complex (phospholipoprotein) synthesized by alveolar cells, with a key role in maintaining respiratory function. There are at least four types of surfactant proteins in vertebrates, i.e., SFTPA, SFTPB, SFTPC, and SFTPD, comprising 8–10% of the surfactant [40,41,42,43,44]. SFTPs are classified into two groups according to their hydrophobicity properties. Hydrophobic proteins include SFTPB and SFTPC, which are primarily involved in the prevention of alveolar collapse [45], while hydrophilic SFTPA and SFTPD belong to the C-type lectin family (collectins), characterized by an N-terminal collagen-like domain and a C-terminal carbohydrate recognition domain that allows binding to various types of macromolecules, pathogens, and allergens, and plays a vital role in host defense [42,46]. In this study, the transcription of SFTPA1-like, SFTPA2-like, SFTPB, SFTPC, and SFTPD-like were identified in *M. fissipes* (Figure 6).

SFTPA is one of the most abundant proteins in pulmonary surfactant and is involved in both host defense and surfactant-related functions. SFTPA has been identified in most vertebrates, including lungfish, frogs, chickens, mice, monkeys, humans, etc. It showed high homology with 64% of the amino acid sequence being conserved in these species overall [44]. It has been generally considered that SFTPA is encoded by a single gene in most species, except in primates and humans, which have two functional genes (Sftpa1 and Sftpa2) [47]. Several previous studies have identified functional differences between SFTPA1 and SFTPA2 in a variety of innate immunity and surfactant-related functions, including cytokine production [48,49], modulation of surfactant secretion [50], and phagocytosis by alveolar macrophages [51,52]. With the increase in genome data, two coding genes of SFTPA (Sftpa1 and Sftpa2) were also identified in amphibian species (e.g., *X. tropicalis*) [53] and avian species, such as *Gallus domesticus* [54], *Alectura lathami*, *Ceuthmochares aereus*, etc. [55], while functional research is lacking. In this study, two coding genes of SFTPA were identified in the lungs of *M. fissipes* and the expression of SFTPA1-like and SFTPA2-like showed opposite variation trends with the progression of metamorphosis. This might be an important aspect of coping with the shift of pathogen groups after transition from aquatic to air breathing.

SFTPB is the most ancient member of the SFTP family and is widely present in species from bony fishes to tetrapods. The Sftpb gene tends to be more broadly expressed in multiple tissues of ray-finned fishes and lungfish, while it is specifically expressed in the lungs of tetrapods [56]. It rearranges lipid molecules in the fluid lining of the lung so that the alveoli can more easily inflate [43], and its absence inevitably leads to lung conditions, with the most common being acute respiratory distress syndrome [57]. Consistent with its significance to air breathing, its transcription in the lungs increased with the maturation of *M. fissipes*.

SFTPC is a genetic innovation for air breathing in vertebrates [56]. It is a hydrophobic membrane protein that increases the rate at which surfactant spreads over the surface, and is a requirement for the proper biophysical function of the lungs [58,59,60]. Humans and animals that are born lacking SFTPC tend to develop progressive interstitial pneumonitis [61]. In *M. fissipes* lungs, the transcriptional level of SFTPC, despite fluctuating, was maintained at a considerable level across the larval and adult stages (Figure 6), suggesting that SFTPC was constitutive of lung development and functionalization.

Similar to SFTPA, SFTPD plays important roles in the host defense against infectious microorganisms and in regulating innate immune response to a variety of pathogen-associated molecular patterns [62]. Interestingly, its expression suddenly dropped to extremely low levels after the metamorphic climax phase in the *M. fissipes* lung. It is possible that there are other proteins, or SFTPD subtypes, that substitute their immune roles in lungs, or that the pathogens, specifically targeted by SFTPD, decrease/vanish after the metamorphosis climax. In any case, our results suggested that the lungs of amphibians might experience functional remodeling during the metamorphic climax phase to achieve adaptation to air breathing.

HB is the protein responsible for oxygen transport in the red blood cells of most vertebrates. Although these proteins were not expressed in pneumonocytes, their richness in the lungs is a significant determinant of gas exchange efficiency. In all jawed vertebrates, erythrocytes produced at distinct developmental stages contain different forms of hemoglobin [63]. Most species produce embryonic-specific hemoglobin in primitive erythroid cells from the yolk sac, some species produce a fetal-specific form in the liver, and all species make an adult hemoglobin in the erythroid cells in bone marrow [63,64,65]. HB is a heterotetramer that consists of two α-like globins and two β-like globins [18,66]. In humans, the α-like globin cluster contains the embryonic ξ gene, and adult α1 and α2 genes, and the β-like globin cluster includes the embryonic ε gene, the fetal ^G^γ and ^A^γ, and adult β and δ genes. During early embryonic development, erythropoiesis transitions from yolk sac to fetal liver, and the site of erythropoiesis is the bone marrow after birth. Accordingly, two switches occur in the expression of genes from the globin cluster during these processes [63,67]. In *X. laevis*, two clusters of globin genes switch from larval type to adult type during metamorphosis [68,69]. A total of 13 HB subunit genes were identified from the *X. laevis* genome, including 8 α-like globin genes (i.e., hba1, hba2, hba3, hba4, hba3-like, hba5-like, hbz, and hbe) and 5 β-like globin genes (i.e., hbb2, hbb2-like, hbg1, hbg2, and hbd) [70]. In this study, we identified seven globin genes in *M. fissipes*, including three β-like globin genes and four α-like globin genes. The adult-type HB subunits replaced the larval-type HB subunits during the metamorphic climax phase, which was considered to be an adaptation strategy for the change of respiratory medium from water to air (Figure 6). This might be coupled with the shift of the respiratory organ from the gill to the skin and lungs. HB switching, coupled with the maturation of an air-breathing organ, is worthy of further study.

### 4.5. Completely Functional Maturation in Adult Lungs

*Microhyla fissipes* adults experience a series of complex life activities such as courtship, reproduction, and hibernation, and they face diverse, complicated, and risky terrestrial environments. Air-breathing ability is a key factor for adapting to a terrestrial environment. Well-functioning and structured lungs improve respiratory efficiency and avoid the risk of environmental pathogens entering from the air during gas exchange. The lungs are fully developed in adults of *M. fissipes*. The WGCNA analysis indicated that the genes related to regulating tissue growth and differentiation (e.g., WNT, BMP, and Notch signaling pathways) increased their transcriptional activity (Figure 2C and Figure 5). It has been evidenced that signaling interplay between BMPs and Notch, WNT, and Hippo signaling cascades co-regulate endothelial cell biology, of which BMP signaling is a core signaling cascade in the endothelium during vascular development and homeostasis [71]. In addition, signaling pathways activate upregulated transcription of pulmonary circulation-related genes (e.g., angiogenesis and platelet production and activation) (Figure 5). Angiopoietin, angiomotin, endothelin, and vascular endothelial growth factor D play crucial roles in angiogenesis [72,73,74]. Their coordinated expression implies that the morphogenesis of blood vessels occurs during this period. Thrombocytes, nucleated hemostatic blood cells of amphibians, are regarded as the functional equivalent of enucleated mammalian platelets, which are involved in hemostasis and immunity [75]. The enhanced function of pulmonary circulation contributes to meeting higher ventilatory demand. However, increased ventilation frequency might give rise to higher immune pressure. Accordingly, the genes involved in immunity (innate immunity and adaptive immunity) presented a robust transcriptional upregulation (Figure 5), which guaranteed the respiratory process. All in all, the growth- and development-related signal pathways were active in adult individuals, which facilitated structurally and functionally complete lungs for adapting to complex terrestrial environments.

### 4.6. Evolutionary Inspiration from Development of M. fissipes Lung

Rapid changes in the morphology and function of *M. fissipes*’ lungs during metamorphosis facilitate realization of air breathing. At the aquatic life stage, the larval lungs are ventilated (Figure 1B), which contributes positive buoyancy and facilitates larval locomotion in water [76]. Meanwhile, the lungs possess a simple blood–gas barrier lacking the alveolar septum with few capillaries (Figure 1C,D), which suggests low respiratory efficiency in the lungs at this stage. Thus, histologically and functionally, larval lungs are more like the swim bladders of fish at the aquatic life stage. Conversely, at the terrestrial life stage, *M. fissipes*’ lungs are highly vascularized and possess robust immunity, which means high efficiency of respiration at this stage. It has been shown that the lungs and swim bladders are homologous organs [56]. The development of the *M. fissipes*’ lungs might recapitulate the processes of air-breathing organs’ evolution from water to land. Therefore, to some extent, revealing the structural construction and molecular switches of lungs during metamorphosis provides clues to the evolution of the lung in vertebrates.

## 5. Conclusions

In this study, our results highlighted the four key steps of lung development for land dwelling: rapid tissue growth during the pro-metamorphic phase, structural construction and molecular switches during the metamorphic climax phase, increased smooth muscle contraction during the pro-metamorphic phase, and further morphogenesis and improved immunity in adults. This study highlights the role of lung development during metamorphosis in facilitating the transition of amphibians from water to land.

## Figures and Tables

**Figure 1 biology-11-00528-f001:**
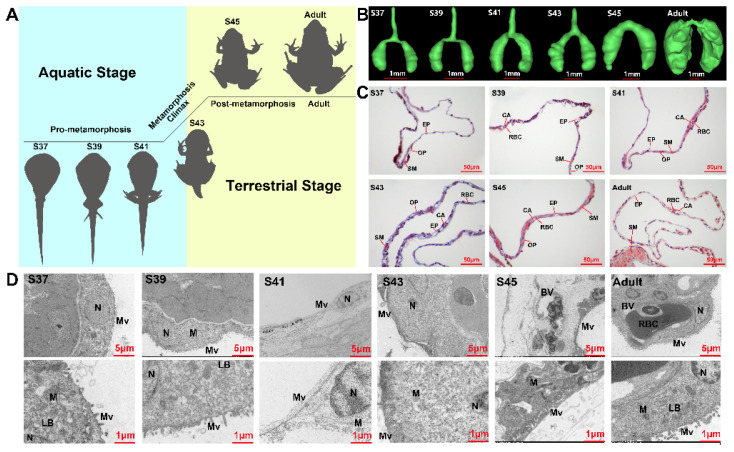
Structural variations of lung with developmental stages. (**A**) Experimental design for this study. Six developmental stages comprising four developmental phases (pro-metamorphic, metamorphic climax, post-metamorphic, and adult phases) with their corresponding habitats. (**B**) Morphological appearance form under micro-CT in different developmental stages (*n* = 2 each stage). (**C**) Histological and (**D**) ultrastructural characteristics of the lung in different developmental stages (*n* = 3 each stage): CA, capillary; EP, epithelium; RBC, red blood cell; SM, smooth muscle; OP, outer pleura. LB, lamellar body; M, mitochondria; Mv, microvillus; N, nucleus.

**Figure 2 biology-11-00528-f002:**
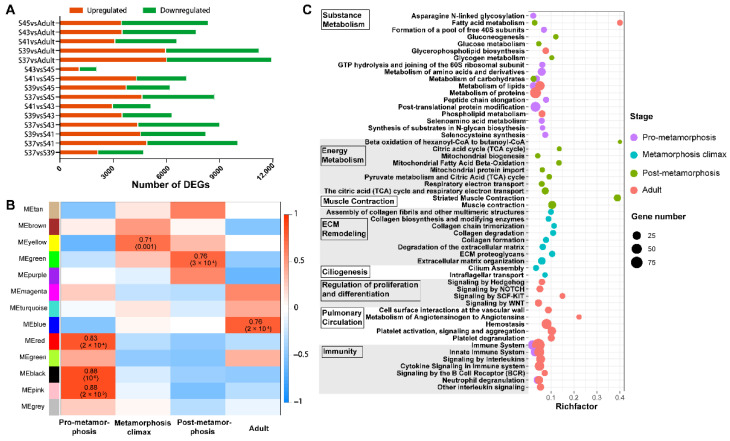
WGCNA and functional enrichment of gene modules. (**A**) The DEGs identified in all pairwise comparisons of stages; (**B**) The correlations between module eigengenes and developmental phases. The color scale indicates the strength of correlation; (**C**) The main functional items of enriched gene modules (FDR < 0.05) in different developmental stages (*n* = 3 each stage).

**Figure 3 biology-11-00528-f003:**
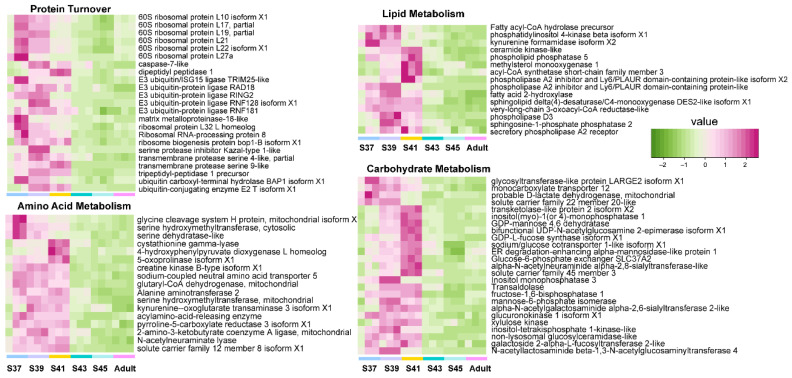
The variation patterns of featured genes involved in protein turnover, lipid metabolism, amino acid metabolism, and carbohydrate metabolism in the pro-metamorphic phase (*n* = 3 each stage).

**Figure 4 biology-11-00528-f004:**
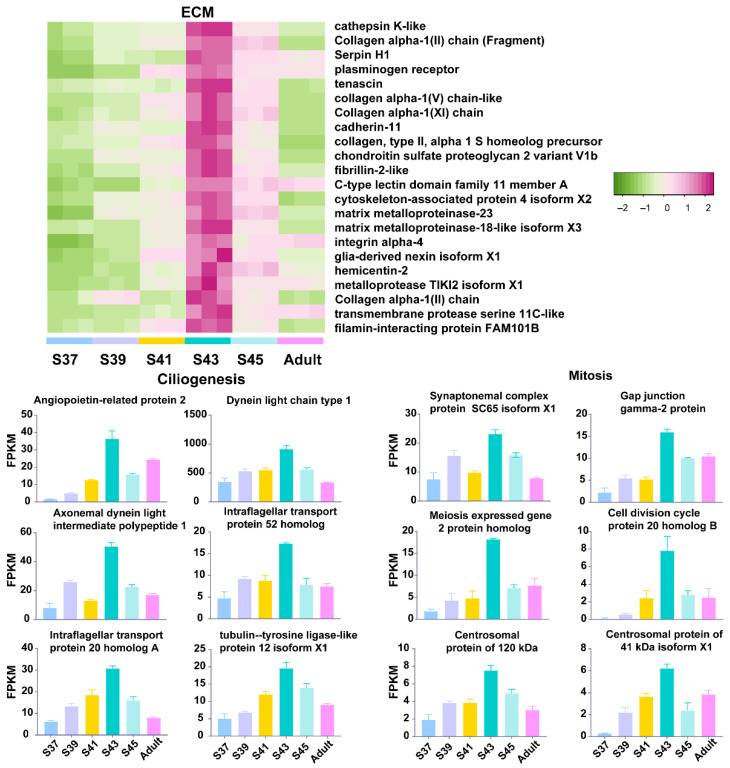
The variation patterns of featured genes involved in ECM construction, ciliogenesis and mitosis in the metamorphic climax phase (*n* = 3 each stage).

**Figure 5 biology-11-00528-f005:**
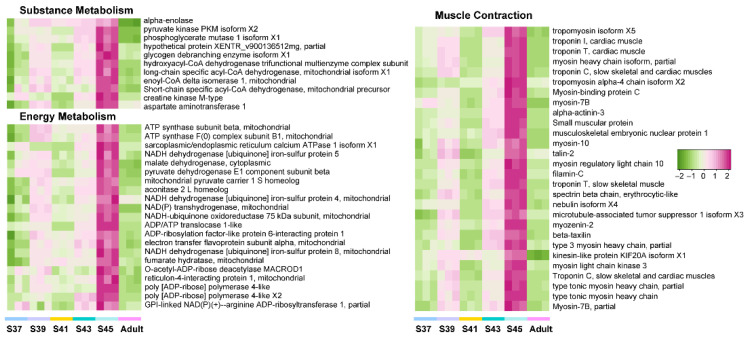
The variation patterns of featured genes involved in substrate metabolism, energy metabolism, and muscle contraction in the post-metamorphic phase (*n* = 3 each stage).

**Figure 6 biology-11-00528-f006:**
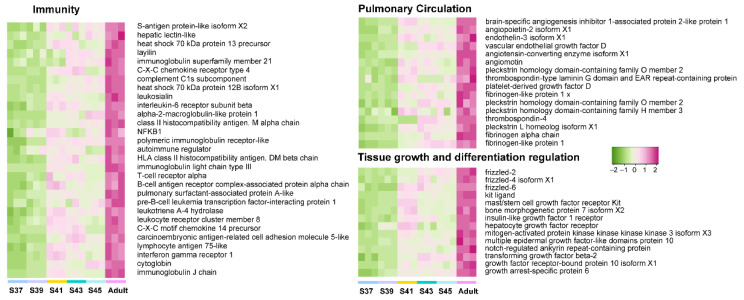
The variation patterns of featured genes involved in immunity, pulmonary circulation, and tissue growth and differentiation regulation in the adult phase (*n* = 3 each stage).

**Figure 7 biology-11-00528-f007:**
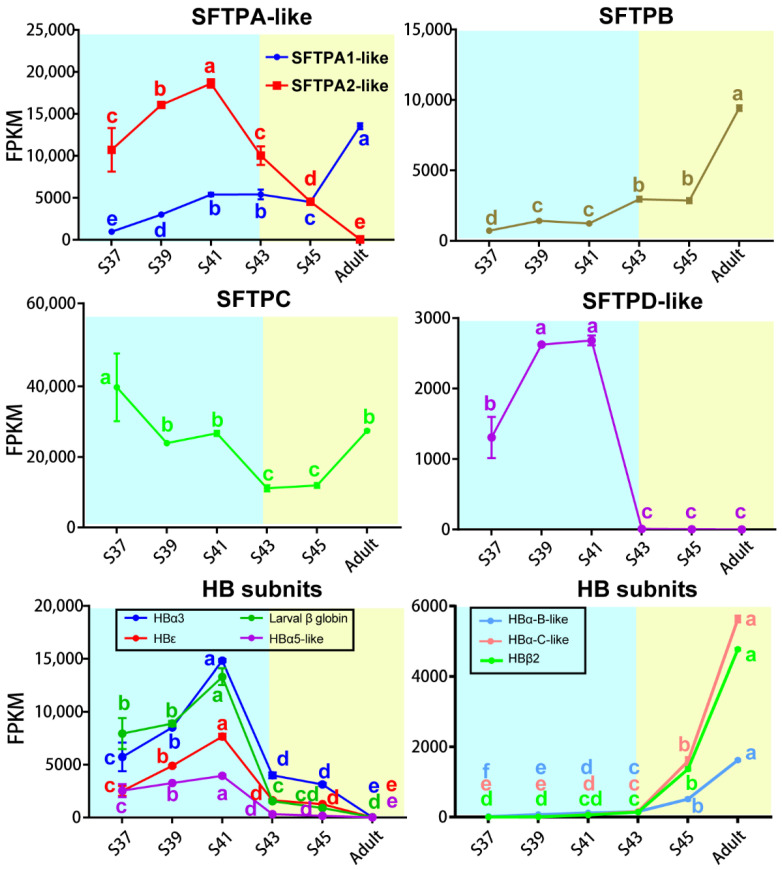
Transcriptional variation patterns of SFTPs and HBs subunits in the lung. Blue and yellow backgrounds represent the aquatic and terrestrial phases, respectively (*n* = 3 each stage), Different letters denote significance between groups at a threshold of *p* < 0.05 (one-way ANOVA and S-N-K post-hoc test).

## Data Availability

The sequencing data in this study have been submitted to the Genome Sequence Archive (GSA; https://bigd.big.ac.cn/gsa/ (accessed on 1 January 2022)) under accession number PRJCA004230.

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
