# Peer review of "From Water to Land: The Structural Construction and Molecular Switches in Lungs during Metamorphosis of *Microhyla fissipes"

_biology, 2022, doi:10.3390/biology11040528_

Round 1

Reviewer 1 Report

The paper characterizes the structural variations of lung with developmental stages, and describes the functional enrichment of gene modules by WGCNA. The finding is of general interest to evolutional developmental biology of amphibians.

I suggest that  the authors rephrase some part of the introduction sections, such as mentioning of “One of the most interesting question is how the different organ systems are organized to function at different developmental stages in 69 matching with the shift of respiration medium from water to air”. The current study still cannot answer this, and other important questions raised by the authors in the introduction part. Focusing on the results would be more appropriate.

Author Response

Dear Reviewers:

Here we would like to submit our revised manuscript entitled “From Water to Land: The Structural Construction and Molecular Switches in Lungs during Metamorphosis of Microhyla fissipes”, MS ID#: biology-1582553. We really appreciate the valuable comments of the reviewers. We have thoroughly modified the manuscript according to their suggestions and used the editing services of MDPI, ID#: english-41657. Details of the revisions have been provided below.

Reviewer 1 Comments and Suggestions for Authors

  • The paper characterizes the structural variations of lung with developmental stages, and describes the functional enrichment of gene modules by WGCNA. The finding is of general interest to evolutional developmental biology of amphibians.

Reply: Thanks for your helpful comments. We followed your suggestion to modify the issues in the manuscript.

  • I suggest that the authors rephrase some part of the introduction sections, such as mentioning of “One of the most interesting question is how the different organ systems are organized to function at different developmental stages in 69 matching with the shift of respiration medium from water to air”. The current study still cannot answer this, and other important questions raised by the authors in the introduction part. Focusing on the results would be more appropriate.

Reply: Thanks for your helpful comments. We have rephrased the introduction sections to focuse on our results (lines 73-75, 86-88)

Reviewer 2 Comments and Suggestions for Authors

  • The authors used integrated approaches to study the structural and molecular changes of amphibian lungs during metamorphosis. The results extended our understanding on the molecular processes and their coordination that underpin the transition of amphibians from aquatic to terrestrial life stages. Moreover, they found some pulmonic-specific molecular markers whose expression levels shifted during the metamorphic climax. These genes might be implicated in the evolution of the air respiration in tetrapod. Overall, this paper provides vast quantities of data, and the statistical analyses in this study are reliable. I think the results are interesting to the readers, and thus recommend it for publication after minor revision.

Reply: Thanks for your helpful comments. We followed your suggestion to modify the issues in the manuscript and answered your questions.

Detailed comments:

  • Line 32: “was” to “is”;

Reply: Done. Thanks.

  • Lines 61: “were” to “are”; "adapted” to “adapt”;

Reply: Done. Thanks.

  • Lines 212-216: Elucidated the criteria for selecting the parameter threshold of module membership and gene significance.

Reply: We determined the parameter threshold of module membership (MM) and gene significance (GS) through multiple tentative calculations, and each of them were conforms to standard. We finally selected the optimal parameter threshold used in this study.

  • Lines 250-258: This paragraph is repetitive and should be deleted;

Reply: Done. Thanks.

  • Lines 264: “R2” to “R2”;

Reply: Done. Thanks.

  • Lines 308-309: The legend of Figure 5 is missed.

Reply: We have added the legend of Figure 5 (lines 308-309).

  • Lines 453-454: Since the type of pathogens likely vary with the respiration medium, do you think is it possible that the pathogens, SFTPD specifically targeted, are decreased/vanished after metamorphosis climax?

Reply: Yes, it is a reasonable assumption. We have added it in new manuscript (lines 446-447).

  • A suggestion for authors: this work finds some interesting molecular switches at transcriptional level. It is supposed to explore the protein switches and conduct functional verification experiment in further work.

Reply: Thanks for your useful suggestion. We will consider these work in our further work.

Reviewer 3 Comments and Suggestions for Authors

  • Paper: "From Water to Land: The Structural Construction and Molecular Switches in Lungs during Metamorphosis of Microhyla fissipes" is covering important and interesting topic to field of zoology.

Reply: Thanks for your helpful comments. We followed your suggestion to modify the issues in the manuscript.

I just have some minor suggestions on paper

  • Try to change sentences which are same and repeated through Simple summary, Abstract and Introduction (lines 11-12, 27-28, 46-49, 60).

Reply: We rewrote the simple summary (lines 11-17) and revised related sentences in abstract (lines 34-37).

  • You can write that skin is important in gas exchange during all life stages of anurans.

Reply: We followed your suggestion to modify the sentence in the manuscript (lines 70-71).

  • In introduction a role of changes in oxidative stress and reactive oxygen species in developing lungs during metamorphosis can be mentioned (Oxidative stress in Pelophylax esculentus complex frogs in the wild during transition from aquatic to terrestrial life and Oxidative stress, tissue remodeling and regression during amphibian metamorphosis)

Reply: We added related introduction in the manuscript (lines 77-78).

  • In Discussion delete parts referring on figures and tables (e.g. line 336, 337, 343 and other) as we already read them in Result section.

Reply: We followed your suggestion to modify the issues in the manuscript (lines 336-338, 342-343, 359-360, 384-385, 479-481).

In addition, we modified the citation format based on MDPI format.

Reviewer 2 Report

The authors used integrated approaches to study the structural and molecular changes of amphibian lungs during metamorphosis. The results extended our understanding on the molecular processes and their coordination that underpin the transition of amphibians from aquatic to terrestrial life stages. Moreover, they found some pulmonic-specific molecular markers whose expression levels shifted during the metamorphic climax. These genes might be implicated in the evolution of the air respiration in tetrapod. Overall, this paper provides vast quantities of data, and the statistical analyses in this study are reliable. I think the results are interesting to the readers, and thus recommend it for publication after minor revision.

Detailed comments:

Line 32: “was” to “is”;

Line 61: “were” to “are”; "adapted” to “adapt”;

Line 212-216: Elucidated the criteria for selecting the parameter threshold of module membership and gene significance.

Line 250-258: This paragraph is repetitive and should be deleted;

Line 264: “R2” to “R2”;

Line 308-309: The legend of Figure 5 is missed.

Line453-454: Since the type of pathogens likely vary with the respiration medium, do you think is it possible that the pathogens, SFTPD specifically targeted, are decreased/vanished after metamorphosis climax?

A suggestion for authors: this work finds some interesting molecular switches at transcriptional level. It is supposed to explore the protein switches and conduct functional verification experiment in further work.

Author Response

(The authors gave the same response as above.)

Reviewer 3 Report

This article explores the transcriptional and physiological changes in the lungs of metamorphosing M. fissipesfrogs.

Please find my comments and suggestions below.

Some of the text is in red. I am not sure if this is a revised and resubmitted manuscript but since there is no authors rebuttal accompanying the MS, I suggest that the authors remove the red font.

Parts of the MS, such as the abstract need to be edited for grammar.

The images in Figure 1C and D are rather low resolution. It would be great to be able to see these images in larger size and with greater resolution. How many tissues were examined for each stage and are represented by the images in Fig. 1?

Were the animals or their lungs perfused to remove circulating blood cells before RNA seq analyses? Amphibian red blood cells are nucleated and express many genes, which are undoubtedly subject to change with metamorphosis. Circulating leukocytes will likewise possibly change their gene expression with transition to and through metamorphosis. Thus, the presented data may not be strictly representative of lung tissues.

It reference to Figure 2, it is not clear how many animals’ lungs were examined per developmental stage. What is the N per group here? If the authors used lungs from individual animals per group, they cannot conclude that differentially expressed genes are due to developmental differences rather than animal to animal variation. Fig. 2B is not informative while the text in 2C needs to be bigger.

Lines 258: ribosomal components and proteolysis-related genes: please elaborate as cellular protein turnover is associated with proteosome functions and not ribosomes or specific proteolytic enzymes. When I look over the list of genes denoted as involved in protein turnover in Fig. 3, I see many genes that do not belong in this category. In fact, most of these genes are not associated with cellular protein turnover. I have less knowledge about the other categories, but I recommend that the authors reevaluate these and confirm their lists.

Throughout the materials section and for every legend, please indicate the N per group.

I believe the term is pre-metamorphic, not pro-metamorphic.

Figure 4 is illegible. Please increase font size and image resolution.

The data in Figure 7 require statistical analysis and denotation. What is the N?

Author Response

Comments and Suggestions for Authors

This article explores the transcriptional and physiological changes in the lungs of metamorphosing M. fissipes frogs.

Please find my comments and suggestions below.

Reply: Thanks for your helpful comments. We followed your suggestion to modify the issues in the manuscript and answered your question.

  • Some of the text is in red. I am not sure if this is a revised and resubmitted manuscript but since there is no authors rebuttal accompanying the MS, I suggest that the authors remove the red font.

Reply: Yes, this is a revised manuscript. We have removed the red font in the new version.

  • Parts of the MS, such as the abstract need to be edited for grammar.

Reply: Thanks for your suggestion. The manuscript has been edited used the editing services of MDPI, ID#: english-41657. The certificate is attached.

  • The images in Figure 1C and D are rather low resolution. It would be great to be able to see these images in larger size and with greater resolution.

Reply: The figures were compressed in manuscript. We also provided the Figures as independent files in the attachment.

  • How many tissues were examined for each stage and are represented by the images in Fig. 1?

Reply: Three tissues were examined for each stage for histological sections (both light and electron microscopy,N=3 each stage) and two tissues were examined for each stage for Micro-computed tomography (N=2 each stage). Thanks for your suggestion, we have added related information in the methods.

  • Were the animals or their lungs perfused to remove circulating blood cells before RNA seq analyses? Amphibian red blood cells are nucleated and express many genes, which are undoubtedly subject to change with metamorphosis. Circulating leukocytes will likewise possibly change their gene expression with transition to and through metamorphosis. Thus, the presented data may not be strictly representative of lung tissues.

Reply: We didn’t perfuse to remove circulating blood cells from lung tissues. The red blood cells and circulating leukocytes of the amphibians express many genes, and their transcriptional profiles vary with the proceeding of metamorphosis. However, our main target is to study the structural and molecular changes in lungs associated with the maturation of the air-breathing. The primary function of blood cells is to deliver the oxygen, while the lung is the site for gas exchange. The numbers and gene expression patterns of the blood cells in the lungs are closely associated with the respiration function of the lung. The gene expression patterns of circulating leukocytes can reflect the immune functions in lungs. Although our data may not be strictly representative of lung tissues, we think the overall changes in gene expression profiles in the lungs (including the circulating cells) would present more comprehensive view on the functional maturation of this respiration organ. Thanks for your comments!

  • It reference to Figure 2, it is not clear how many animals’ lungs were examined per developmental stage. What is the N per group here? If the authors used lungs from individual animals per group, they cannot conclude that differentially expressed genes are due to developmental differences rather than animal to animal variation. 2B is not informative while the text in 2C needs to be bigger.

Reply: Mixed sample sequencing is effective and generally accepted method to solve the problem that sample is tiny as long as the samples from similar biological states. In this study, all the mixed samples are form same population and same developmental stages. Therefore, the transcriptional result can present the features of developmental stage.

In detail, 30, 30, 20, 15, 10 and 1 individual lungs were merged as one sample at stages 37, 39, 41, 43, 45, and adult, respectively; three samples (biological replicates) were prepared for each stage. We have provided more detailed sampling protocol in methods.

Fig. 2B presented the processes finding the modules of co-expressed genes each developmental phase including pro-metamorphosis (stage 37, 39, and 41), metamorphosis climax (stage 43), pro-metamorphosis (stage 45), and adult. Therefore, it is better to keep the figure.

We provided the figures of high resolution in the attachment.

  • Lines 258: ribosomal components and proteolysis-related genes: please elaborate as cellular protein turnover is associated with proteosome functions and not ribosomes or specific proteolytic enzymes. When I look over the list of genes denoted as involved in protein turnover in Fig. 3, I see many genes that do not belong in this category. In fact, most of these genes are not associated with cellular protein turnover. I have less knowledge about the other categories, but I recommend that the authors reevaluate these and confirm their lists.

Reply: Protein turnover is the balance between protein synthesis and protein degradation. More synthesis than breakdown indicates an anabolic state that builds lean tissues, more breakdown than synthesis indicates a catabolic state that burns lean tissues (ToyamaandHetzer, 2013). The ribosome is the site of protein synthesis. The transcriptional level of ribosomal components can suggest the ability of protein synthesis and the transcriptional level of proteolysis-related genes can suggest the ability of protein degradation. Therefore, the transcriptional level of ribosomal components and proteolysis-related genes can suggest the ability of protein turnover (Chang et al., 2021; Wang et al., 2019). When we reevaluate the gene list, we find some misplaced genes. We followed your suggestion and modified the figure. Thanks for your important question!

  • Throughout the materials section and for every legend, please indicate the N per group.

Reply: Thanks for your suggestion, we have added related information in methods and legend.

  • I believe the term is pre-metamorphic, not pro-metamorphic.

Reply: I disagree with the comment. Metamorphosis can be divided into three periods based on the sequence of developmental processes: pre-metamorphosis, where tadpoles grow and develop hindlimb buds; pro-metamorphosis, where hindlimbs develop and digits differentiate; and metamorphic climax, where forelimbs emerge and the tail is resorbed completely (Liu et al., 2018). And at stage 37, the toes have elongated and webbed. Two hindlimbs could be seen from the dorsal part in M. fissipes (Figure 1A) (Wang et al., 2017). In this study, pro-metamorphic phase includes stage 37, 39 and 41. Therefore, the term is “the pro-metamorphic”, not “pre-metamorphic”.

  • Figure 4 is illegible. Please increase font size and image resolution.

Reply: Thanks for your suggestion. We have increased the font size and provided high-resolution figures as independent files.

  • The data in Figure 7 require statistical analysis and denotation. What is the N?

Reply: Thanks for your suggestion. We followed your suggestion and conducted statistical analysis for the data in Figure 7.

Reference:

  1. Chang, L., Wang, B., Zhang, M., Liu, J., Zhao, T., Zhu, W., Jiang, J. (2021). The effects of corticosterone and background colour on tadpole physiological plasticity. Comparative Biochemistry and Physiology Part D: Genomics and Proteomics 39: 100872
  2. Liu, L., Zhu, W., Liu, J., Wang, S., Jiang, J. (2018). Identification and differential regulation of micrornas during thyroid hormone-dependent metamorphosis in microhyla fissipes. BMC Genomics 19: 507
  3. Toyama, B.H., Hetzer, M.W. (2013). Protein homeostasis: Live long, won't prosper. Nature Reviews Molecular Cell Biology 14: 55-61
  4. Wang, S., Zhao, L., Liu, L., Yang, D., Khatiwada, J.R., Wang, B., Jiang, J. (2017). A complete embryonic developmental table of microhyla fissipes (amphibia, anura, microhylidae). Asian Herpetological Research 8: 108-117
  5. Wang, X., Chang, L., Zhao, T., Liu, L., Zhang, M., Li, C., Xie, F., Jiang, J., Zhu, W. (2019). Metabolic switch in energy metabolism mediates the sublethal effects induced by glyphosate-based herbicide on tadpoles of a farmland frog microhyla fissipes. Ecotoxicology and Environmental Safety 186: 109794

Reviewer 4 Report

Paper:"From Water to Land: The Structural Construction and Molecular Switches in Lungs during Metamorphosis of Microhyla fissipes" is covering important and interesting topic to field of zoology. 

I just have some minor suggestions on paper

Try to change sentences which are same and repeated through Simple summary, Abstract and Introduction (lines 11-12, 27-28, 46-49, 60)

You can write that skin is important in gas exchange during all life stages of anurans. In introduction a role of changes in oxidative stress and reactive oxygen species in developing lungs during metamorphosis can be mentioned (Oxidative stress in Pelophylax esculentus complex frogs in the wild during transition from aquatic to terrestrial life and Oxidative stress, tissue remodeling and regression during amphibian metamorphosis)

In Discussion delete parts referring on figures and tables (e.g. line 336, 337, 343 and other) as we already read them in Result section.

Author Response

(The authors gave the same response as above.)

Round 2

Reviewer 3 Report

I feel that the authors adequately addressed my concerns